# Therapeutic Targeting of DNA Repair Pathways in Pediatric Extracranial Solid Tumors: Current State and Implications for Immunotherapy

**DOI:** 10.3390/cancers16091648

**Published:** 2024-04-25

**Authors:** Sophia J. Zhao, Daniel Prior, Christine M. Heske, Juan C. Vasquez

**Affiliations:** 1Department of Pediatric Hematology/Oncology, Yale University School of Medicine, New Haven, CT 06510, USA; sophia.j.zhao@yale.edu (S.J.Z.); daniel.prior@yale.edu (D.P.); 2Pediatric Oncology Branch, National Cancer Institute, National Institutes of Health, Bethesda, MD 20892, USA; christine.heske@nih.gov

**Keywords:** DNA damage response, immune checkpoint inhibition, synthetic lethality, pediatric extracranial solid tumors, neuroblastoma, osteosarcoma, Ewing sarcoma, rhabdomyosarcoma

## Abstract

**Simple Summary:**

Survival for many pediatric cancers has improved over recent decades. However, for pediatric patients with solid tumors that fail to respond to standard therapies, or relapse after initial response, outcomes generally remain poor, indicating a need for novel and improved treatments. Many cancers have an impaired ability to repair DNA damage, which in excess can become toxic to cells. As such, one potential approach for these challenging cancers is to target the DNA damage repair pathways of cancer cells, with the goal of inducing a lethal amount of DNA damage. This article reviews the current research efforts into targeting DNA damage repair pathways in pediatric extracranial solid tumors. It reviews the biology of DNA damage repair pathways, the biology of several extracranial pediatric cancers, the preclinical research investigating targeting the DNA damage repair in pediatric cancers, and the clinical trials using these agents in patients. This article also reviews the ability to harness a patient’s immune system to kill cancer cells, and the research that has been done investigating ways in which DNA damage can activate the anti-tumor immune response.

**Abstract:**

DNA damage is fundamental to tumorigenesis, and the inability to repair DNA damage is a hallmark of many human cancers. DNA is repaired via the DNA damage repair (DDR) apparatus, which includes five major pathways. DDR deficiencies in cancers give rise to potential therapeutic targets, as cancers harboring DDR deficiencies become increasingly dependent on alternative DDR pathways for survival. In this review, we summarize the DDR apparatus, and examine the current state of research efforts focused on identifying vulnerabilities in DDR pathways that can be therapeutically exploited in pediatric extracranial solid tumors. We assess the potential for synergistic combinations of different DDR inhibitors as well as combinations of DDR inhibitors with chemotherapy. Lastly, we discuss the immunomodulatory implications of targeting DDR pathways and the potential for using DDR inhibitors to enhance tumor immunogenicity, with the goal of improving the response to immune checkpoint blockade in pediatric solid tumors. We review the ongoing and future research into DDR in pediatric tumors and the subsequent pediatric clinical trials that will be critical to further elucidate the efficacy of the approaches targeting DDR.

## 1. Introduction

DNA damage is detected and repaired via numerous intra- and inter-cellular signaling events and enzymes, which collectively comprise the apparatus known as the DNA damage response (DDR) [1]. Upon detection of DNA damage, the DDR system leads to cell-cycle arrest, regulation of DNA replication, and DNA repair [2]. If DNA repair is not possible, DDR can affect downstream cell fate decisions, leading to either cell senescence or apoptosis via various mechanisms [2]. The DDR apparatus comprises five major pathways, including base excision repair (BER), nucleotide excision repair (NER), mismatch repair (MMR), homologous recombination repair (HRR), and non-homologous end joining (NHEJ). Over 450 proteins are thought to be integral to DDR function [2].

DDR dysfunction is a hallmark of tumorigenesis and cancer in humans. Deficiencies in DDR pathways lead to genomic instability, generating clonal heterogeneity and oncogene activation and/or loss of tumor suppressor genes, collectively promoting tumorigenesis [3]. The importance of DDR dysfunction in promoting tumor formation is evidenced by the fact that germline DDR gene mutations underlie many cancer predisposition syndromes and are common somatic mutations in a variety of tumors [3].

Therapeutically, DDR deficiencies represent potential vulnerabilities for cancer cells, which, in order to prevent excess genomic instability, may depend on compensatory DDR pathways to survive [1,3]. This dependency on alternative DDR pathways presents an opportunity for synthetic lethal killing of tumor cells by targeting compensatory DDR pathways [2]. An example of the potential of such synthetic lethality is the clinical efficacy of poly (ADP-ribose) polymerase (PARP) inhibitors in the treatment of BRCA-deficient tumors, which has prompted investigations into other molecular targets in DDR pathways [4,5].

In this review, we will examine the current state of research efforts focused on identifying vulnerabilities in DDR pathways that can be therapeutically exploited in pediatric malignancies. We will also assess the potential for synergistic combinations of different DDR inhibitors as well as combinations of DDR inhibitors with chemotherapy. Lastly, we will consider the immunomodulatory implications of targeting DDR pathways and the potential for using DDR inhibitors to enhance tumor immunogenicity and improve response to immune checkpoint blockade in immunologically “cold” pediatric solid tumors.

## 2. Overview of DNA Damage Repair Pathways

### 2.1. Base Excision Repair

BER is the DDR pathway that is primarily responsible for single-strand DNA (ssDNA) break repair as well as for correcting small DNA lesions that do not significantly alter the DNA structure. These lesions typically result from deamination, oxidation, and methylation and can occur due to the spontaneous degradation of DNA as well as external damage caused by chemicals and radiation [6,7]. The first step in BER involves DNA glycosylase, which detects and excises the damaged base. An endonuclease, AP endonuclease 1 (APE1), and an exonuclease then process the excision site, DNA polymerase β inserts the missing nucleotide, and the new nucleotide is sealed by a DNA ligase [7,8]. Several proteins involved in the BER pathway are able to be targeted, including PARP1/PARP2, APE1, and DNA polymerase β [9].

The BER pathway plays a prominent role in responding to alkylating agents and topoisomerase I poisons, two classes of agents that feature prominently in current research targeting DDR pathways. DNA damage induced by alkylating agents can be repaired via two pathways: PARP-dependent BER or by the direct removal of O6-methyl guanines by the DNA repair enzyme O6-methlguanine-DNA methyltransferase (MGMT) [10]. Topoisomerase I poisons, including irinotecan, induce stalling of the topoisomerase I complex, which leads to the formation of ssDNA breaks. These breaks are then processed by PARP1, in conjunction with the DNA repair enzymes tyrosyl-DNA phosphodiesterase 1 (TDP1) and polynucleotide kinase/phosphatase (PNKP), and ultimately repaired [10].

### 2.2. Nucleotide Excision Repair

NER is responsible for resolving bulky DNA lesions commonly induced by intrastrand cross-links caused by alkylating chemotherapeutic agents, environmental carcinogens, and ultraviolet radiation. After recognition of the lesion, the helicases xeroderma pigmentosum groups B (XPB) and D (XPD) unwind the DNA, and replication protein A (RPA) and xeroderma pigmentosum groups A (XPA) and G (XPG) are recruited to form the xeroderma pigmentosum group C (XPC) protein complex, which coordinates excision and repair [11].

Preclinical studies targeting several enzymes within the NER pathway, including excision repair cross-complementation group 1/xeroderma pigmentosum group F (ERCC1/XPF) and ERCC1/XPA, have demonstrated anti-tumor activity [12,13]. Currently, however, there are no drugs targeting the NER pathway in clinical trials.

### 2.3. Mismatch Repair

MMR corrects base mismatches, insertions, and deletions that are generated during DNA replication [14]. Eukaryotic MutS homologs (MSHs) recognize the mismatches and insertion or deletion mispairs. The MSHs then recruit eukaryotic MutL homologs (MLHs), which triggers an incision of the mismatch by an exonuclease. The gap in DNA is then re-synthesized by a DNA polymerase and ligated by a DNA ligase [15]. MMR is also responsible for replication errors within microsatellite regions of DNA, and MMR deficiency can lead to microsatellite instability (MSI).

The MMR pathway has been found to be mutated in various cancers, and MMR-deficient cancers have been shown to be immunogenic tumors, given the high rates of formation and expression of non-self-neoantigens [16]. As such, MMR-deficient tumors have been found to be particularly sensitive to immune checkpoint inhibition, so much so that the U.S. Food and Drug Administration (FDA) approved the programmed cell death protein 1 (PD-1) inhibitor, pembrolizumab, for unresectable, MMR-deficient, unresponsive solid tumors; this was notably the very first tumor, age, and site agnostic, biomarker-driver approval [17]. While specific MMR proteins are not currently targetable, immunotherapeutic approaches for MMR-deficient tumors continue to be actively investigated.

### 2.4. Homologous Recombination

Homologous recombination repair (HRR) is the main mechanism for the repair of double-strand break (DSB) lesions that occur in the S and G2 phases of the cell cycle. HRR relies on utilizing the sister chromatid as a template, thereby repairing DNA damage in an error-free manner [18]. After DSBs are recognized by the MRN complex (consisting of Mre11, Rad50, and Nbs1), they are processed to create a single-strand DNA overhang at each end. The ssDNA overhangs are coated and stabilized by RPA, followed by the binding of RAD51 with the cooperation of the BRCA1-PALB2-BRCA2 complex. Then, one ssDNA overhang invades a homologous DNA sequence on a sister chromatid and DNA polymerase extends the end of the invading 3’ strand until it can capture and resolve the second ssDNA overhang [19].

Components of HRR are also involved in the repair of interstrand cross-links (ICLs), which are caused by alkylating and platinum-based chemotherapies. ICLs are recognized by a core complex composed of Fanconi Anemia proteins, which in turn serve to recruit ICL repair proteins and ultimately lead to RAD51-mediated HRR of DSBs [20]. Several proteins in the HRR pathway are targetable, including Ataxia telangiectasia-mutated (ATM), Ataxia telangiectasia and Rad3-related proteins (ATR), and checkpoint kinase 1 and 2 (CHK1/2).

### 2.5. Non-Homologous End Joining

NHEJ is another integral pathway responsible for repairing DSBs that can occur throughout any phase of the cell cycle. As opposed to HRR, which relies on an undamaged template, NHEJ directly re-ligates the two broken DNA strands and thus is more prone to errors [21]. Upon cell detection of a DSB, the Ku heterodimer is recruited to the broken DNA strands. Subsequently, the DNA-dependent protein kinase catalytic subunit (DNA-PKcs) is recruited and forms a complex with Ku to help further stabilize the DNA ends. Then, any residual damaged or overhanging DNA segments are processed, followed by ligation [22]. Several proteins within the NHEJ pathway are targetable, including ATM and DNA-dependent protein kinase (DNA-PK).

## 3. Clinically Targetable DNA Damage Repair Proteins in Cancer

Synthetic lethality-based treatment strategies for cancers with underlying DDR defects are an area of active investigation. Here we will review the DDR proteins that are currently able to be targeted clinically.

### 3.1. ATM

The *Ataxia telangiectasia mutated* gene (*ATM*) encodes a serine-threonine protein kinase that is a main transducer of target proteins involved in the DNA DSB repair pathways NHEJ and HRR [23,24]. Once activated, ATM phosphorylates a variety of downstream target proteins, including CHK2, which, in turn, phosphorylates various substrates that induce cell-cycle arrest and initiate DNA repair processes [25].

ATM deficiency or ATM inhibition can induce synthetic lethality in cancer cells that harbor other underlying DDR deficiencies. For example, cells with loss-of-function mutations in the HRR genes *BRCA1*/*2* as well as *ATM* struggle to repair DSBs, leading to synthetic lethality [26]. Similarly, hypermethylation of the *ATM* promoter region can result in ATM deficiency, resulting in impaired DDR [25,27]. ATM is inactivated in approximately 5% of all cancers but is estimated to be inactivated in a larger proportion of mantle cell lymphomas and colorectal and uterine cancers [28]. In light of this, pharmacologic inhibitors of ATM are being explored as potential cancer treatments, especially in combination with DNA-damaging agents like chemotherapy and radiation [25].

### 3.2. ATR

ATR is a serine/threonine-specific protein kinase primarily activated when ssDNA regions are detected, resulting in increased replication stress [29]. Replication stress is the general term that describes the stresses that result in altered replication fork progression, decreased replication accuracy, and DNA breaks [30]. ATR plays a central role in responding to replication stress, phosphorylating a wide array of target proteins, one of the most important being CHK1. Activated CHK1 phosphorylates a variety of downstream substrates involved in coordinating DDR [31]. ATR plays a crucial role in various DDR pathways, including HRR. In particular, ATR activates key HRR proteins such as BRCA1 and RAD51 [32].

ATR synthetic lethality is observed in cancer cells with certain DDR deficiencies. For example, inhibition of ATR in ATM-deficient cells results in the accumulation of DSBs, which cannot be repaired due to the dysfunction of ATM and CHK2; this ultimately results in cell death [33]. DNA damaging agents, such as temozolomide (TMZ), can also lead to activation of the ATR/CHK1 pathway, resulting in a synergistic interaction with pharmacological ATR inhibitors [34,35].

### 3.3. CHK1/2

As mentioned earlier, CHK1 is the major downstream effector of ATR and prevents cells with DNA damage from entering into mitosis [36]. Once phosphorylated by ATR, CHK1 triggers the S- and G2/M-phase checkpoints [36]. In response to DNA damage, ATM activates CHK2, which mediates the G1/S cell-cycle checkpoint via p53 [37]. Inhibition of CHK1 allows cells with unrepaired DNA damage to enter mitosis, subsequently undergoing apoptosis due to incompletely replicated chromosomes [38].

Thus, pharmacological targeting of CHK1 has been studied as a means of inducing tumor cell death. Preclinical data have demonstrated the anti-tumor activity of the CHK1 inhibitor prexasertib as both monotherapy and in combination with PARP inhibitors and cytotoxic chemotherapy agents [39,40]. CHK1/2 inhibition remains an area of active investigation. Recent and ongoing trials continue to evaluate the efficacy of novel CHK1 inhibitors, such as SRA737 and BBI-355, as monotherapy and in combinations [41,42].

### 3.4. PARP

PARPs are a family of enzymes that transfer ADP-ribose to target proteins [43]. The PARP1 protein primarily plays a role in the detection and repair of DNA single-strand breaks (SSBs) [44]. Upon detection of an SSB, PARP1 becomes activated and then creates poly (ADP-ribose) (PAR) chains. These PAR chains serve as signals that attract various DDR proteins, including X-ray repair cross-complementing protein 1 (XRCC1), to the site of DNA damage [45]. In addition, it has been suggested that PARP1 may also be involved in NHEJ and HRR [46].

PARP inhibitors (PARPi) are known to induce synthetic lethality in cells with HRR deficiency, such as those with *BRCA1/2* mutations. Preclinically, pharmacological inhibition of PARP1 causes DNA replication fork collapse, which would normally be repaired by the HRR pathway. In cells with *BRCA1/2* mutations and thus impaired HRR, the use of PARPi leads to an inability to repair the collapsed DNA replication forks and synthetic lethality [47]. Tumors can also display a BRCAness phenotype in which they do not have *BRCA1/2* mutations but instead harbor mutations in other DDR genes, such as *ATR* and *ATM*, or mutations in Krebs cycle genes, such as *IDH1/2*, that result in increased sensitivity to PARPi [48,49,50,51]. In tumors with Krebs cycle mutations, there is an accumulation of oncometabolites, such as 2-hydroxyglutarate (2HG), succinate, and fumarate. One proposed mechanism for greater PARPi sensitivity involves oncometabolite-induced inhibition of lysine demethylases, which in turn leads to histone hypermethylation at loci surrounding DNA breaks, masking a local H3K9 trimethylation signal involved in the proper recruiting of homologous recombination proteins [49]. Alternatively, 2HG accumulation has been associated with an increase in heterochromatin and higher levels of replication stress that is dependent on PARP for repair [52].

### 3.5. WEE1

The protein kinase Wee1 is an inhibitory regulator of the G2/M cell-cycle checkpoint [53]. In a normal G2/M transition, polo-like kinase 1 (PLK-1) phosphorylates Wee1, marking Wee1 for degradation, which allows the cell to proceed through mitosis [53]. When DNA damage is present, the ATM/ATR pathways negatively regulate PLK-1, thus stabilizing Wee1, which inhibits CDK1 and leads to G2 arrest that allows for DNA repair [54]. In tumors with a DDR deficiency, Wee1 inhibition is thought to abrogate the G2/M checkpoint, leading these cells to undergo mitosis and synthetic lethal cell death [53]. The *TP53* tumor suppressor gene, which codes for the p53 protein, plays a key role in regulating the G1/S checkpoint, and as such, *TP53*-mutated cells are largely reliant on the G2 checkpoint for survival. These factors collectively make Wee1 inhibition a potential target in *TP53*-mutated cancers [53,55]. This mechanism has been supported by the selective efficacy of Wee1 inhibition in multiple *TP53*-mutated preclinical models, including in breast cancer, non-small cell lung cancer, and glioblastoma [53,56,57]. It should be noted that Wee1 inhibition has also demonstrated efficacy in various cancer cell lines independent of p53 function [58]. Of note, the anti-tumor activity of Wee1 inhibition can be counteracted by tumor overexpression of Myt1, a kinase that also regulates the G2/M checkpoint and has somewhat overlapping activity with Wee1 [59]. In the clinical setting, many clinical trials investigating the efficacy of Wee1 inhibition in various contexts have been conducted, with several demonstrating anti-tumor activity [60].

### 3.6. DNA-PK

DNA-PK is a serine-threonine protein kinase complex composed of the DNA-PK catalytic subunit (DNA-PKcs) and a heterodimer of Ku proteins, Ku70/Ku80 [61]. The main role of DNA-PK in DDR is to repair DNA DSB via NHEJ [62,63]. DNA-PKcs is dysregulated in multiple cancers, including chronic lymphomas, colorectal, prostate, breast, and brain cancers [63]. As such, DNA-PK has emerged as a therapeutic target in malignancy.

DNA-PK is currently being studied in the preclinical and early-phase clinical trial settings. Preclinical models have shown efficacy in DNA-PK inhibition in sensitizing cancer cells to chemotherapy and radiotherapy [64,65]. Early-phase clinical trials are underway investigating DNA-PK inhibition for a variety of advanced tumors [63].

## 4. Targeting DNA Damage Repair Pathways in Pediatric Cancers

### 4.1. Neuroblastoma

Neuroblastoma (NB) originates from neural crest progenitor cells and constitutes the most common extracranial solid tumor in infants and children [66]. Patients with high-risk diseases continue to have inferior outcomes despite intensive multimodal therapies, with a five-year overall survival rate of roughly 50% [66].

Approximately 20–30% of high-risk NB are characterized by hemizygous deletion of chromosome bands of 11q22-q23, which include the *ATM* locus [67]. One study demonstrated that approximately 36% (16/45) of examined NB-derived cell lines were ATM-deficient [63], with another study finding *ATM* loss in 28% (14/50) of NB patient samples [67]. *ATM* loss in human NB cell lines has been shown to correlate with increased tumor formation and growth [67]. In addition to *ATM* loss, NB can harbor an overexpression of CHK1 [68].

ATM-deficient NB cell lines and xenograft models exhibit increased sensitivity to PARPi [69,70]. In addition, pharmacological inhibition of CHK1 [71,72] and Wee1 have both been shown to reduce cellular proliferation in some NB models, an effect that is potentiated when they are combined [68]. A follow-up study in NB xenografts demonstrated that the Wee1 inhibitor adavosertib was minimally efficacious as a single agent but exhibited anti-tumor activity when combined with irinotecan [73]. Further research is needed to investigate the effects of other DDR inhibitors in *ATM*-deficient NB models.

A pediatric phase I study of the Wee1 inhibitor adavosertib plus irinotecan in children with relapsed solid and CNS tumors identified a recommended phase II dose (RP2D) and included two patients with NB, one of whom had stable disease (SD) (Table 1) [74]. In a follow-up phase II expansion cohort, three out of 20 patients with NB (15%) demonstrated an objective response, meeting the study defined efficacy endpoint and suggesting this combination may warrant future investigation [75]. The European ESMART trial treated 20 patients with recurrent/refractory NB with adavosertib and carboplatin. Two patients had a partial response (PR), also suggesting that Wee1 may be the preferred target for NB [76]. The ADVL1411 Children’s Oncology Group (COG) phase I/II trial of the PARPi talazoparib in combination with low-dose TMZ in children included two patients with NB, one of whom had SD [77]. Another phase I trial studying the combination of talazoparib and irinotecan with and without TMZ in pediatric patients with recurrent or refractory solid tumors included a single patient with NB who had SD [78], again suggesting there may be a benefit of targeting PARP as part of a drug combination in a subset of patients with NB.

### 4.2. Osteosarcoma

Osteosarcoma (OS) is a malignancy of mesenchymal origin and the most common primary malignant bone tumor in adolescents [81,82]. While localized disease is often curable, patients with metastatic disease have a poor prognosis with a 5-year overall survival rate of <30% despite the standard of care multi-agent chemotherapy and surgical resection [81,83].

OS frequently carries genomic alterations associated with sensitivity to DDR inhibition [84]. *TP53* is the most frequently mutated gene in OS, with both alleles estimated to be mutated in 80–100% of tumors, suggesting that *TP53* mutations are key drivers of tumorigenesis in OS [85]. Chen et al. performed whole-genome sequencing of OS tumor samples from 19 patients and found p53 pathway lesions in 100% of these tumors [85]. Given the frequency of *TP53* mutations, OS cells are often reliant on G2/M arrest in order to repair DNA damage, making DDR targeting an attractive potential therapeutic approach. Another study showed that approximately 80% of OS samples displayed a genomic signature characteristic of BRCA1/2 deficient tumors [86]. Additionally, a decreased expression of alpha-thalassemia/mental retardation, X-linked (ATRX), a protein involved in the alternative lengthening of telomeres, is common in OS and has been associated with increased sensitivity to ATR inhibition in other tumor types [87,88]. *ATR* has been found to be overexpressed in OS, supporting this potential mechanism and therapeutic approach [89].

Preclinically, DDR targeting in OS has yielded promising results. OS cell lines have shown sensitivity to the PARPi talazoparib alone and in combination with current standard-of-care therapies for OS [86]. However, in a separate high-throughput drug screen, a majority of OS cell lines did not show PARPi sensitivity when compared to BRCA1 breast tumor models, including the OS models with previously determined genomic profiles consistent with a BRCAness phenotype [90]. A follow-up study found an association between HRR deficiency with talazoparib sensitivity in OS cell lines [91]. OS cell lines have also shown sensitivity to ATR inhibition [89] and CHK1/2 inhibition [72,92]. Wee1 inhibition as monotherapy has been shown to induce OS cell death [93] as well as increase the radiosensitivity of OS cells, which is classically considered to be radioresistant [94]. Wee1 inhibition has been found to synergistically reduce OS cell viability when combined with ATR inhibition [95] as well as with gemcitabine [93].

Clinical results with monotherapy targeting DDR for patients with OS have been disappointing thus far (Table 1). The COG phase II trial of the Wee1 inhibitor adavosertib with irinotecan included three patients with recurrent OS, none of whom responded to this combination [74]. The COG phase I study investigating the CHK1/2 inhibitor prexasertib included two patients with OS, neither of whom responded [38]. The COG ADVL1411 phase I/II clinical trial of the PARPi talazoparib in combination with low-dose TMZ in children included four patients with OS, none of whom had a response [77]. Another phase I trial studied the combination of talazoparib and irinotecan included three patients with recurrent/refractory OS, two of whom had SD, suggesting there may be modest activity with this combination [78]. There is currently an ongoing phase II trial studying the combination of the PARP inhibitor olaparib with the ATR inhibitor ceralasertib in patients with recurrent OS [84].

### 4.3. Ewing Sarcoma

Ewing sarcoma (ES) is the second most common bone cancer among children and can arise from the bone or soft tissue [96,97]. Survival outcomes for metastatic and relapsed disease remain low despite intensive multimodal therapy consisting of chemotherapy, surgical resection, and radiotherapy [98,99]. A majority of ES tumors possess fusions between two genes, *Ewing sarcoma breakpoint region 1* (*EWSR1*) and *Friend leukemia integration 1* (*FLI1*) [100,101]. The resulting EWS-FLI1 fusion protein acts as an aberrant transcription factor that activates or represses target genes, thus promoting oncogenesis [102]. It has been shown that EWS-FLI1 binds to *EWSR1* and downregulates its activity. This results in decreased HRR and the accumulation of R-loops, nucleic acid structures composed of a DNA-RNA hybrid and the non-template DNA strand, thereby increasing replication stress [30]. The increased replication stress and decreased HRR characteristic of ES have made targeting DDR pathways a potentially appealing therapeutic approach.

Much of the work targeting DDR in ES thus far has been related to targeting PARP. Brenner et al. determined that EWS-FLI1 drives the expression of PARP1, which acts in a positive feedback loop by further promoting EWS-FLI1-mediated transcription [101]. The EWS-FLI1 fusion protein has also been shown to positively regulate the expression of Schlafen family member 11 protein (SLFN11), a DNA/RNA helicase that is recruited during replication stress and induces cell death [103,104,105]. SLFN11 inhibits replication and causes prolonged replication fork stalling during the S phase of mitosis, thereby enhancing sensitivity to PARPi [106].

Brenner et al. demonstrated that ES cell lines with EWS-FLI1 fusions were sensitive to PARPi, as they potentiated greater DNA damage due to the abrogation of multiple PARP1-driven DDR pathways [101]. In preclinical ES murine models, PARPi was efficacious in improving survival when combined with irinotecan and TMZ but not when used as monotherapy; the combination of PARPi with chemotherapy-induced durable and complete remissions in a majority of mice [107]. Multiple other preclinical studies have similarly demonstrated that PARPi sensitize ES models to TMZ [101,108,109,110,111,112] as well as ionizing radiation [113] and other therapeutic agents [114,115,116].

In addition to PARP inhibitors, inhibitors of ATR, Wee1, CHK1, DDK, and DNA-PK have demonstrated preclinical activity in ES models. ATR inhibition has shown single-agent activity against ES cell lines in vitro, as well as in in vivo xenografts [117]. A recent study also demonstrated the synergy between ATRi and cisplatin in ES cell lines as well as in ES xenografts [118]. ES cells have demonstrated a particular susceptibility to the inhibition of ribonucleotide reductase, the rate-limiting enzyme in deoxyribonucleotide synthesis [119]. This finding has led to multiple investigations, which have demonstrated the ES susceptibility in in vitro and in vivo models to the combined inhibition of ribonucleotide reductase and either Wee1, ATR, or CHK1 [93,119,120,121,122]. The combination of Wee1 and PARP inhibition has also shown efficacy in ES cell lines [123], as has the combination of DNA-PK inhibition with PARP inhibition [124]. Independent of Wee1 inhibition, DNA-PK inhibitors when used in combination with topoisomerase 2 poisons, such as etoposide or doxorubicin, have demonstrated synergy in in vitro and in vivo models of ES [125]. Another target that has shown promise in ES cells is DBF4-dependent kinase (DDK), a serine/threonine kinase with multiple cellular functions including the activation of the cellular response to replication stress. Preclinical targeting of DDK, using the DDKi XL413 and TAK-931, has been shown to reduce ES cell line viability, both as monotherapy and in combination with Wee1 inhibition [126,127].

Clinical responses to PARPi monotherapy have been largely disappointing, while combination therapy has yielded more promising results (Table 1). A phase II trial testing the PARP inhibitor olaparib as a single agent included 22 adult patients with advanced ES. Of the 12 evaluable patients, no objective responses were seen, although four experienced SD [79]. The first evaluation of a PARP inhibitor plus chemotherapy in pediatric patients with relapsed/refractory solid tumors included ES patients. This trial enrolled 16 patients with ES on the talazoparib plus irinotecan arm; among this cohort, the overall response rate was 12.5%, as one patient had a PR and one had a complete response (CR). Nine patients had SD. Seven ES patients were enrolled in the talazoparib plus irinotecan plus TMZ arm, which achieved an overall response rate of 42.9%; three patients had PR, suggesting clinical benefit. An additional two patients in this arm had SD [77]. The COG ADVL1411 phase I/II clinical trial of the PARPi talazoparib in combination with low-dose TMZ included ten pediatric patients with ES, two of whom had SD [75]. Another phase I trial investigated the combination of the PARPi niraparib plus TMZ or irinotecan. Among the 12 patients in the niraparib plus irinotecan arm, there was one PR and six patients with SD [80]. Additional early-phase trials are actively investigating combination therapy of PARP inhibitors with irinotecan, TMZ, or both for patients with advanced ES [98].

In addition to the PARPi trials, other DDR-targeting trials enrolling patients with EWS include the COG ADVL1312, a phase II trial of the Wee1 inhibitor adavosertib with irinotecan. This trial included four patients with advanced ES, with one patient having a PR [74]. In the COG phase I study investigating the CHK1/2 inhibitor prexasertib in pediatric patients with recurrent or refractory tumor patients, a single patient had ES but had progressive disease [38]. There is currently an open phase II study investigating the efficacy of the CHK1 inhibitor, LY2880070, combined with gemcitabine for relapsed/refractory ES cases [128].

### 4.4. Rhabdomyosarcoma

Rhabdomyosarcoma (RMS) originates from undifferentiated mesenchymal cells and is the most common soft tissue sarcoma among children [129]. Despite multimodal therapy including chemotherapy, surgical resection, and radiotherapy, the outcomes for patients with metastatic disease remain dismal [129,130]. RMS is classified into three major subtypes: (1) tumors that harbor pathogenic fusion proteins between paired box gene 3 (PAX3) or PAX7 and forkhead box O1 (FOXO1), which are predominantly of the histologic alveolar subtype; (2) tumors bearing mutations in the *myogenic differentiation 1* (*MYOD1*) gene, which are predominantly of the spindle cell/sclerosing subtype; and (3) tumors with neither of these lesions, which are predominantly of the embryonal subtype [130,131,132]. RMS tumors that are *FOXO1* fusion-positive are more frequently metastatic and chemotherapy-resistant than *FOXO1* fusion-negative tumors [133]. *MYOD1* mutant tumors also carry poor outcomes, with one recent retrospective analysis demonstrating a 4-year survival rate of only 18% in *MYOD1*-mutated RMS [134].

The biology of RMS makes DDR targeting an attractive potential therapeutic approach. The expression of PAX3-FOXO1 has been shown to increase tumor replication stress and increase reliance on the ATR/CHK1 repair pathway, making ATR pathway molecules potential therapeutic targets [135,136]. PARP levels have also been demonstrated to be elevated in both *FOXO1* fusion-positive and fusion-negative RMS cell lines, suggesting that PARP inhibition may have a role in the treatment of RMS [137]. In a multi-omics characterization of RMS, *Wee1* was found to be more highly expressed at the mRNA level in RMS relative to other pediatric cancers with resultant dysregulation of the G2/M pathway [138]. It has also been shown that RMS cell lines highly express human TDP1, an enzyme that repairs stalled topoisomerase I-DNA complexes [139,140].

In preclinical models, *FOXO1* fusion-positive RMS has shown increased sensitivity to ATR inhibition, suggesting this may be a promising target in this subset of tumors [135]. Preclinical studies have also identified that PARP inhibition is efficacious in RMS. Yan et al. demonstrated the efficacy of combined olaparib plus TMZ against both embryonal and alveolar RMS in zebrafish and mouse models, whereas single-agent PARP inhibition was ineffective [141]. Fam et al. demonstrated the single-agent activity of both TDP1 and PARP inhibition in RMS cell lines as well as the combined efficacy of either TDP1 or PARP inhibition with irinotecan analogues [139]. In another preclinical study, the PARP inhibitor talazoparib was found to be the most effective when combined with SN-38, the active metabolite of irinotecan, in RMS cell lines [142]. Additional preclinical work demonstrated that the Wee1 inhibitor AZD1775, alone and in combination with irinotecan or vincristine, led to G2/M phase arrest, increased DNA damage, and had anti-tumor activity against in vivo models of high-risk RMS [138,143]. Several studies have also demonstrated that RMS cells with increased levels of TDP1 have chromosomal instability and are highly sensitive to inhibitors of histone deacetylases (HDACs), presumably from alterations in the epigenetic regulation of DDR [140,144]. A recent preclinical trial in cell lines and patient-derived xenograft models of alveolar RMS demonstrated the striking single-agent anti-tumor effect of the ATRi elimusertib, suggesting that this may be a promising approach in the future [145].

Clinically, the efficacy of DDR targeting in RMS remains largely unknown to date as few patients with RMS have been enrolled in relevant trials (Table 1). In a COG phase I study investigating the CHK1/2 inhibitor prexasertib in pediatric patients with recurrent or refractory tumor patients, four patients had RMS, and all had progressive disease [38]. In the COG ADVL1411 trial of the PARP inhibitor talazoparib, with low-dose TMZ, a single patient with RMS was enrolled and had progressive disease [77]. Another phase I trial investigating the combination of talazoparib and irinotecan in pediatric patients with recurrent or refractory solid tumors included three patients with RMS, all of whom had disease progression [78]. PARP inhibition continues to be actively studied in a phase I trial of olaparib with TMZ for patients with advanced ES and RMS [146].

## 5. Rational Drug Combinations with DNA Damage Repair Inhibitors

There are ongoing efforts to identify rational DDR inhibitor-based combinations to enhance synthetic lethality (Table 2). While there have been several promising DDRi combinations identified in preclinical studies that are now moving to early-phase clinical trials in adult patients, this approach has not been extensively studied in pediatric cancers outside of the PARPi and TMZ combination studies reviewed above [147,148]. Moreover, significant dose-limiting toxicities, namely myelosuppression, remain a barrier to the wide applicability of DDRi combination approaches [77,78,148].

In the pediatric population, the recently completed AcSé-ESMART trial combining the ATRi ceralasertib and the PARPi olaparib showed that this combination was well-tolerated with evidence of anti-tumor activity in patients with refractory/relapsed advanced solid tumors. Efficacy was seen in tumors that demonstrated molecular alterations consistent with HRR deficiency or replication stress. This trial included 18 patients with a variety of solid tumors (eight sarcomas, five central nervous system tumors, four neuroblastomas, and one carcinoma). One patient with pinealoblastoma demonstrated a PR, while another patient with neuroblastoma had prolonged SD that later converted to a PR [162]. More research is needed to uncover DDRi combinations that are effective and well-tolerated in pediatric patients.

## 6. Targeting the DNA Damage Response and Immune Checkpoint Blockade

Tumors commonly exploit homeostatic inhibitory immune checkpoints, such as cytotoxic T lymphocyte-associated protein 4 (CTLA-4) and PD-1, to suppress T cell effector function and escape immune surveillance [163]. While immune checkpoint blockade (ICB) has shown activity in the adult population, clinical trials in pediatric patients have generally yielded disappointing results, with the exception of Hodgkin lymphoma [164].

Putative predictive biomarkers of ICB response include a high tumor mutational burden (TMB), an increased number of tumor-infiltrating lymphocytes (TILs), an inflammatory gene signature, positive programmed death-ligand 1 (PD-L1) expression, and MMR deficiency/microsatellite instability [165,166]. It has been hypothesized that the limited efficacy of ICB in childhood cancers is due to intrinsic differences in the immunogenicity of tumors between adults and children, with pediatric tumors generally being immunologically “cold” and harboring a lower mutational burden [164,167,168].

In the COG ADVL1412 phase I/II study of single-agent nivolumab, no anti-tumor activity was noted in pediatric patients with recurrent or refractory solid tumors (Table 3) [169]. KEYNOTE-051 was a phase I/II trial evaluating the PD-1 inhibitor pembrolizumab in pediatric patients with melanoma or PD-L1-positive relapsed/refractory solid tumors. Among patients with solid tumors, only 8 of 106 patients achieved a PR [170]. The study concluded that PD-L1 expression alone was not a sufficient means of predicting PD-1 checkpoint inhibitor responsiveness among pediatric solid tumor patients [170]. The COG ADVL1412 phase I/II study also included an arm to investigate the use of nivolumab plus ipilimumab in recurrent/refractory pediatric solid tumors with similarly low response rates, with just 2 of 55 patients having PR, and another 4 patients having SD [169].

Targeting the DNA damage response has garnered significant attention as a potential avenue for inducing immunogenicity and sensitizing “cold” tumors to ICB. DDR defects and/or DDR inhibitors have been shown to remodel the tumor microenvironment and synergize with ICB through DNA damage-induced activation of immune recognition pathways and increased neoantigen formation [171,172]. This combination of DDR inhibitors with immunotherapies is being investigated in pediatric tumors as well [173,174].

PARPi-induced DSBs result in the generation of cytosolic DNA, which is detected by cGMP-AMP synthase (cGAS), leading to activation of the stimulator of interferon genes (STING) pathway. Activation of the STING pathway results in the production of type I interferons (IFNs) and subsequent recruitment of cytotoxic CD8+ T cells [175,176]. Similarly, ATR inhibition has been shown to result in accelerated mitotic entry and increased genomic instability, leading to micronuclei formation, activation of the cGAS/STING pathway, and production of the proinflammatory chemokine CCL5 [177,178]. The immunomodulatory properties of DDR inhibitors have prompted the initiation of several clinical trials, largely in the adult population. The results of these trials have been mixed, with efficacy seen in a subset of patients, particularly those with ovarian and breast cancer where there is clear efficacy for PARPi [179,180,181].

The immune effects of DDR inhibitors in pediatric tumor models have been understudied, and the clinical investigation of DDR inhibitor and ICB combinations in children has lagged significantly behind that in adults. To date, a single phase 1 trial is studying the combination of the PARPi niraparib with the PD-1 inhibitor dostarlimab in pediatric patients with advanced solid tumors [149]. The pediatric clinical investigation into combination DDRi and ICB likely awaits a signal from the adult data, beyond the subsets of patients known to respond to PARPi.

## 7. Conclusions and Future Directions

DNA damage is fundamental to human cancer initiation and progression. Our past and present armory of cytotoxic agents largely rely on further inducing DNA damage in tumor cells, to the point of lethality. As we have learned more about the complex biology of tumors, we have incorporated the targeting of DNA repair. Beyond the well-established synthetic–lethal interaction between PARP inhibitors and *BRCA1/2* mutations, there exist similar interactions between other agents and mutated or silenced DDR genes. Moreover, combination strategies with multiple DDR-targeting treatments, DNA-damaging agents, radiation, and ICB show promise in preclinical settings and the potential for efficacy as therapeutic approaches in clinical settings.

For pediatric patients, the clinical evaluation of drugs targeting DDR pathways has lagged behind their adult counterparts yielding limited, if not disappointing, results thus far. To optimize the design of pediatric trials, eligibility based on the mutational status of key genes, rather than histology, may be a better approach, as molecular profiling efforts have revealed that specific gene mutations are often found across multiple histologies. The evidence to date suggests that the biomarkers in pediatrics for DDR responsiveness greatly differ from those in adult malignancies, highlighting the need for further identification of relevant pediatric biomarkers [182]. Such potential biomarkers include measurements of replication stress (e.g., R-loops), chromosome 11q loss in NB, and aberrant transcription factor gene fusions, among others [182]. As novel agents are developed, such as the relatively recent and promising DNA polymerase theta (PolQ) inhibitors [183], an improved understanding of the nuances of tumor biology and immunobiology is needed to create biology-driven combinations of therapies that will provide the greatest benefits to patients.

An additional clinical challenge facing the study of DDR-targeting agents, both as single agents and in combinations, is drug toxicity. Overlapping toxicities are of particular concern for combining DDR-targeting agents and immunotherapy. The identification of tissue-specific biomarkers, as well as the development of tumor-targeted delivery strategies are key to improving the safety and efficacy of these therapies [182]. Lastly, more research is needed to identify the ideal dosing and scheduling of DDR inhibitors when given with DNA-damaging agents and/or ICB agents.

In summary, the ongoing and future research into DDR in pediatric tumors and the subsequent pediatric clinical trials will be critical to further elucidate the efficacy of the approaches targeting the DDR discussed in this review.

## Figures and Tables

**Table 1 cancers-16-01648-t001:** Outcomes of pediatric solid tumor patients treated with DNA damage response inhibitors.

Disease	Target	Agent	Combination	Patient Population	N	Responses	Phase	Study	Ref.
Neuroblastoma									
	Wee1	Adavosertib	Irinotecan	Relapsed pediatric solid tumors	2	1 SD	I	NCT02095132	[74]
	Wee1	Adavosertib	Irinotecan	Relapsed pediatric solid tumors	20	1 PR, 3 SD	II	NCT02095132	[75]
	PARP	Talazoparib	TMZ	R/R pediatric solid tumors	2	1 SD	I/II	NCT02116777	[77]
	PARP	Talazoparib	Irinotecan	R/R pediatric solid tumors	1	1 SD	I	NCT02392793	[78]
Osteosarcoma									
	Wee1	Adavosertib	Irinotecan	Relapsed pediatric solid tumors	3	None	II	NCT02095132	[74]
	CHK1/2	Prexasertib	N/A	R/R pediatric solid tumors	2	None	I	NCT02808650	[38]
	PARP	Talazoparib	TMZ	R/R pediatric solid tumors	4	None	I/II	NCT02116777	[77]
	PARP	Talazoparib	Irinotecan	R/R pediatric solid tumors	3	2 SD	I	NCT02392793	[78]
Ewing sarcoma									
	PARP	Olaparib	N/A	Adult advanced Ewing sarcoma	12	4 SD	II	NCT01583543	[79]
	PARP	Talazoparib	Irinotecan	R/R pediatric solid tumors	16	1 CR, 1 PR, 9 SD	I	NCT02392793	[78]
	PARP	Talazoparib	Irinotecan + TMZ	R/R pediatric solid tumors	7	3 PR	I	NCT02392793	[78]
	PARP	Talazoparib	TMZ	R/R pediatric solid tumors	10	2 SD	I/II	NCT02116777	[77]
	PARP	Niraparib	Irinotecan	Advanced Ewing sarcoma	12	1 PR, 6 SD	I	NCT02044120	[80]
Rhabdomyosarcoma									
	CHK1/2	Prexasertib	N/A	R/R pediatric solid tumors	4	None	I	NCT02808650	[38]
	PARP	Talazoparib	TMZ	R/R pediatric solid tumors	1	None	I/II	NCT02116777	[77]
	PARP	Talazoparib	Irinotecan	R/R pediatric solid tumors	3	None	I	NCT02392793	[78]

N, Number of patients; SD, stable disease; PR, partial response; CR, complete response; TMZ, temozolamide; R/R, Recurrent/Refractory.

**Table 2 cancers-16-01648-t002:** Active clinical trials targeting DNA damage repair inhibitors in pediatric patients.

Target	Agent	Phase	Combination	Patient Population	Ages	Study	Ref.
PARP	Niraparib	I/II	Dostarlimab	R/R solid tumors	6 months–18 years	NCT04544995	[149]
PARP	Talazoparib	I	Topotecan, Gemcitabine	Relapsed AML	Up to 21 years	NCT05101551	[150]
PARP	Olaparib	II	None	R/R solid tumors, non-Hodgkin lymphoma, Histiocytic disorders with DNA damage repair defects	1–21 years	NCT03233204	[151]
PARP	Talazoparib	I/II	Nanoliposomal irinotecan, TMZ	R/R solid tumors	1–30 years	NCT04901702	[152]
PARP	BGB-290	I	TMZ	IGH ½-mutated gliomas	13–25 years	NCT03749187	[153]
PARP	Veliparib	II	TMZ and radiation	Newly diagnosed gliomas without H3 K27M or BRAFV600 Mutations	3–25 years	NCT03581292	[154]
PARP	Olaparib	I	TMZ	Recurrent Ewing sarcoma or rhabdomyosarcoma	16 years and older	NCT01858168	[146]
PARP	Olaparib	II	Ceralasertib	R/R osteosarcoma	12–40 years	NCT04417062	[155]
ATR	RP-3500 (camonsertib)	I	RP-6306	Locally advanced or metastatic R/R solid tumors	12 years and older	NCT04855656	[156]
ATR	AZD6738	I	Gemcitabine	Locally advanced or metastatic solid tumors	16 years and older	NCT03669601	[157]
ATR	Elimusertib	I/II	None	R/R solid tumors	1–18 years	NCT05071209	[158]
Wee1	Adavosertib	I/II	Irinotecan	R/R solid tumors	1–21 years	NCT02095132	[74]
Wee1	Adavosertib	I	Radiation	Newly diagnosed diffuse intrinsic pontine gliomas	37 months–21 years	NCT01922076	[159]
Wee1	Adavosertib	I/II	Carboplatin	Refractory hematologic or solid tumor	Up to 18 years	NCT02813135	[160]
Wee1	ZN-c3	I/II	Gemcitabine	R/R osteosarcoma	12 years and older	NCT04833582	[161]

TMZ, temozolamide; R/R, Recurrent/Refractory.

**Table 3 cancers-16-01648-t003:** Outcomes of immune checkpoint inhibition therapy in pediatric solid tumor patients.

Trial	Target	Agent	Combination	Patient Population	N	Responses	Phase	Study
COG ADVL1412	PD-1	Pembrolizumab	N/A	R/R pediatric solid tumors	63	None	I/II	NCT02304458
COG ADVL1412	PD-1, CTLA-4	Nivolumab	Ipilimumab	R/R pediatric solid tumors	55	2 PR, 4 SD	I/II	NCT02304458
KEYNOTE-051	PD-1	Pembrolizumab	N/A	Pediatric melanoma, or PD-L1-positive R/R pediatric solid tumors	106	8 PR	I/II	NCT02332668

N, Number of patients; SD, stable disease; PR, partial response; R/R, Recurrent/Refractory.

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
