# Peer review of "Therapeutic Targeting of DNA Repair Pathways in Pediatric Extracranial Solid Tumors: Current State and Implications for Immunotherapy"

_cancers, 2024, doi:10.3390/cancers16091648_

Round 1

Reviewer 1 Report

Comments and Suggestions for Authors

Overall, a readable, succinct and complete review of Therapeutic Targeting of DNA Repair Pathways. The implications for combinations with immunotherapy is a little light- but so is the data.

 1 Introduction

Should probably introduce and define “replication stress”

2.3. Mismatch Repair:

Lines 96-98: Would more carefully spell out that this was the very first tumor agnostic and age agnostic biomarker driven approval.

3.1 ATM

Line 141: The first line of the paragraph is referring to protein, not DNA, so should not be italicized (e.g “ATM deficiency” should be ATM deficiency).

3.2 ATR

Line 160-162: Based on the reference provided (a review) and the evidence discussed therein it is not fair to reduce the p53/ATR story to “Similarly, loss of functional p53, a tumor suppressor protein that is regulated by ATR, results in greater reliance on ATR for cell cycle regulation and increased sensitivity to ATR inhibition [31].” It is context dependent and requires a more nuanced statement. Could change “results in” to “may result in” and/or add at the end, for example, “in some experimental systems, likely dependent on other genomic context.” The other (preferred) option is to leave it out, since it is a complex story (see below- WEE1).

3.3. CHK1/2

Line 173 to 175: When discussing preclinical data one should refer to “antitumor activity” rather than “efficacy.”

Line 180 to 181: Prexasertib is also still in trials – but none have been FDA approved and the future remains unclear.

Overall, This is the only paragraphs in this section of potential therapeutic targets that refers to specific clinical trial results. For other targets, including Parpi which are FDA approved, the discussion is more mechanistic and preclinical in this section. This has the effect of suggesting that CHK1/2 inhibitors  are further along in clinical development, and/or more promising than others. There were single agent responses to ATRi and certainly PARPi as well in biomarker positive breats cancer patients.

3.5. WEE1

Lines 208-281 have a much more nuanced discussion of p53 – and cite primary data as opposed to a review. Would favor leaving it out of ATR ( and could include ATR here).

4.1 Neuroblastoma

Line 250-252 Should state that ph2 expansion of adavosertib actually meet the defined ph2 endpoint for the study. The European eSMART tial also suggests Wee1 is the preferred target for NBL with 2 PR: Gatz SA, et al Phase I/II Study of the WEE1 Inhibitor Adavosertib (AZD1775) in Combination with Carboplatin in Children with Advanced Malignancies: Arm C of the AcSé-ESMART Trial. Clin Cancer Res. 2024 Feb 16;30(4):741-753. PMID: 38051741.

4.2 Osteosarcoma

Should include WEE1 azenosertib trial with gemcitabine here and/or table 2.

4.3 Ewing Sarcoma

Line 316 Replication Stress should be defined/ introduced earlier – see above.

Line 351 the phrase “Concordant with the preclinical data” should be struck. It suggests PARPi in the clinic have been efficacious- to date therapeutic efficacy has been overall disappointing. While intended to mean combination is better than monotherapy, overall, it has not worked which is DIScordant with preclinical data.

Lines 360-367 should clearly state these are single arm studies using known active agents and cannot delineate the contribution of the PARPi to the efficacy, hence the importance of the randomized ONITT trial (NCT04901702)

4.4 Rhabdomyosarcoma

Preclinical discussion should probably include: Pusch FF et al Elimusertib has Antitumor Activity in Preclinical Patient-Derived Pediatric Solid Tumor Models. Mol Cancer Ther. 2024 Apr 2;23(4):507-519. PMID: 38159110; PMCID: PMC10985474.

6. Targeting the DNA Damage Response and Immune Checkpoint Blockade

Probably worth commenting that pediatrics is waiting for a signal from the adult data- especially given that only PARPi have attained FDA approval- and thus far signal for combination therapies has been limited to breast and ovarian, where there is clear efficacy for PARPi. If neither immunotherapy nor DDR inhibition have shown single agent signal, it is highly unlikely that the combination would do so.

7 Conclusions

Lines 506-508 “To optimize the design of pediatric trials, eligibility based on mutational status of key genes, rather than histology, may be a better approach, as molecular profiling efforts have revealed that specific gene mutations are often found across multiple histologies” suggests that the biomarker in peds for DDR will be the same or similar to adults.  The preponderance for evidence, including that cited in this review, point to that that they will not. It would be important to suggest the other potential forms of biomarkers e.g. measurements of replication stress, 11q or others. There is a good discussion in (Pearson ADJ et al. Paediatric Strategy Forum for medicinal product development of DNA damage response pathway inhibitors in children and adolescents with cancer: ACCELERATE in collaboration with the European Medicines Agency with participation of the Food and Drug Administration. Eur J Cancer. 2023 Sep;190:112950. Epub 2023 Jun 21. PMID: 37441939.)

Table 1: Ewing Sarcoma is indented and should come out to the left margin. Table should include the references for the responses

Table 2 should include WEE1 (Azenosertib) trial (NCT04833582) Viswatej Avutu et al., A phase 1/2 dose-escalation and dose-expansion study of ZN-c3 in combination with gemcitabine in adult and pediatric subjects with relapsed or refractory osteosarcoma.. JCO 40, TPS11584-TPS11584(2022).

Table 3: A little off topic, does not really add much.

Reviewer 2 Report

Comments and Suggestions for Authors

This review serves as a good introduction to strategies targeting DNA repair in the treatment of childhood extracranial solid tumors. It covers the potential and current clinical trial picture of combining DNA repair inhibitors with genotoxic agents such as radiation, temozolomide and topoisomerase poisons as well as the application of synthetic lethality. Although the review is well written, there are a few relatively minor clarifications and additions that could be included that would further strengthen the review.

1. In the overview of DNA damage response pathways, the authors should briefly include the direct removal of O6-methyl guanine by MGMT and the pathway (including TDP1 and PNKP) dealing with topoisomerase I poisons because temozolomide and irinotecan feature prominently in current preclinical and clinical research. It would also be worth mentioning that the BER pathway is primarily responsible for single strand break repair as well as base damage.

2. Lines 73-74: It would be preferable to mention the specific exo/endonucleases as well as the specific DNA polymerase commonly used in BER.

3. Lines 81-82: Define abbreviations - XPA = Xeroderma Pigmentosa Group A; RPA = replication protein A

4. Lines 83-84: It should be noted that there are several reports regarding the identification of ERCC1/XPF and ERCC1/XPA. Thus, the statement that “Currently, there are no targetable proteins with the NER pathway” requires some clarification. Do the authors mean there are no drugs targeting NER proteins that are undergoing clinical trials? 

5. Line 108: RPA was already mentioned in line 81. It should be defined there and abbreviated here.

6. Lines 158-160: The sentence “ATM is inactivated….uterine cancers” would be better placed in the ATM section rather than the ATR section.

7. In the ATR section a brief mention should be made of the role of ATR in the response to replication stress.

8. Line 198: The authors mention mutations in Krebs cycle genes increase sensitivity to PARPi. Has a mechanism underlying this observation been discovered? If so, I think a brief explanation of this surprising observation would be useful.

9. In the section on WEE1 it would be good to point out that MYT1 overexpression can counteract the inhibition of WEE1 by Adoversertib (e.g. see Sokhi et al. Front Cell Dev Biol. 2023 Nov 2;11:1270542. doi: 10.3389/fcell.2023.1270542).

10. Lines 235-238: As well as providing the percentages it would be useful to provide the sizes of the cohorts measured.

11. Tables 1 and 2: Add a column for any relevant references.

12. Line 242: CHK1 or CHK2?

13. Lines 281-288: Is there any explanation for the divergent observations regarding OS sensitivity to PARPi?

14. Line 349: What is the reagent used to target DDK?

15. In the Conclusions and Future Directions it might be a good idea to mention Polq inhibitors, which appear to be showing promise.

Reviewer 3 Report

Comments and Suggestions for Authors

Congratulation on this very clear and comprehensive report. The aims and results are very clearly presented, with many reference studies evaluated and analysed. This analysis may be very useful for both clinicians treating patients and for planning the future early phase studies. 

Author Response

We thank reviewer 3 for their positive comments.